# ROBUST REINFORCEMENT LEARNING WITH DISTRIBUTIONAL RISK-AVERSE FORMULATION

## ABSTRACT

The purpose of robust reinforcement learning is to make predictions more robust to changes in the dynamics or rewards of the system. This problem is particularly important when dynamics and rewards of the environment are estimated from the data. However, without constraints, this problem is intractable. In this paper, we approximate the Robust Reinforcement Learning constrained with a $f$-divergence using an approximate Risk-Averse formulation. We show that the classical Reinforcement Learning formulation can be robustified using a standard deviation penalization of the objective. Two algorithms based on Distributional Reinforcement Learning, one for discrete and one for continuous action spaces, are proposed and tested in a classical Gym environment to demonstrate the robustness of the algorithms.

## 1 INTRODUCTION

The classical Reinforcement Learning (RL)Sutton & Barto (2018) problem using Markov Decision Processes (MDPs) modelization gives a practical framework to solve sequential decision problems under uncertainty of the environment. However, for real-world applications, the final chosen policy can sometimes be very sensitive to sampling errors, inaccuracy of the model parameters, and definition of the reward.

This problem motivates robust Reinforcement Learning, aiming to reduce such sensitivity by taking to account that the transition and/or reward function $(P, r)$ may vary arbitrarily inside a given uncertainty set. The optimal solution can be seen as the solution that maximizes a worst-case problem in this uncertainty set or the result of a dynamic zero-sum game where the agent tries to find the best policy under the most adversarial environment (Abdullah et al., 2019). In general, this problem is NP-hard (Wiesemann et al., 2013) due to the complex max-min problem, making it challenging to solve in a discrete state action space and to scale to a continuous state action space.

Many algorithms exist for the tabular case for Robust MDPs with Wasserstein constraints over dynamics and reward such as Yang (2017); Petrik & Russel (2019); Grand-Clément & Kroer (2020a;b) or for $L_\infty$ constrained S-rectangular Robust MDPs (Behzadian et al., 2021). Here we focus on a *more general continuous state space* $\mathcal{S}$ with a discrete or continuous action space $\mathcal{A}$ and with constraints defined using $f$-divergence.

Robust RL (Morimoto & Doya, 2005) with continuous action space focuses on robustness in the dynamics of the system (changes of $P$) and has been studied in Abdullah et al. (2019); Singh et al. (2020); Urpí et al. (2021); Eysenbach & Levine (2021) among others. Eysenbach & Levine (2021) tackles the problem of both reward and transition using Max Entropy RL, whereas the problem of robustness in action noise perturbation is presented in Tessler et al. (2019). Here, we tackle the problem of robustness *through dynamics of the system.*. Recently, the issue of the Robust Q-Learning has also been addressed in Ertefaie et al. (2021).

In this paper, we show that it is possible to tackle a Robust Distributional Reinforcement Learning problem with $f$-divergence constraints by solving a risk-averse RL problem, using a formulation based on mean standard deviation optimization.

The idea beyond that relies on the argument from Robust Learning theory, stating that Robust Learning under an uncertainty set defined with $f$-divergence is asymptotically close to Mean-Variance (Gotoh

et al., 2018) or Mean-Standard deviation optimization (Duchi et al., 2016; Duchi & Namkoong, 2018).

In this work, we focus on the idea that generalization, regularization, and robustness are strongly linked in RL or MDPs as shown in Husain et al. (2021); Derman & Mannor (2020); Derman et al. (2021); Ying et al. (2021); Brekelmans et al. (2022). We show that is it possible to improve the Robustness of RL algorithms with variance/standard deviation regularisation. Moreover, the problem of uncertainty under the distribution of the environment is transformed into a problem with uncertainty over the distribution of the rewards, which makes it tractable.

Note that our work is related to Smirnova et al. (2019) as they penalise the expectation by the variance of returns. However, their approach differs from ours since they use the variance estimate under a Gaussian assumption of distributions while we use a standard deviation penalization without any distribution assumptions. Moreover, the idea of robustness in the change of dynamics is not demonstrated numerically, and the problem tackled is different since they consider close policy distributions, while we consider dynamic distributions.

The contribution of the work is the following: we motivate the use of standard deviation penalization and derive two algorithms for discrete and continuous action space that are robust to changes in dynamics. These algorithms only require one additional parameter tuning, which is the Mean-Standard Deviation trade-off. Moreover, we show that our formulation using Distributional Reinforcement Learning is robust to changing transition dynamics in environments with both discrete and continuous action spaces both in the Mujoco suite and in stochastic environments derived from Mujoco.

**Related topics : Regularised MDPs :** Policy Regularisation in RL Geist et al. (2019) has been studied and led to state-of-the-art algorithms such as PPO and SAC (Schulman et al., 2017b; Haarnoja et al., 2018; Vieillard et al., 2020). In these algorithms, an additional penalisation based on the current policy is added to the classical objective function. The idea is different, as we penalize our mean objective function using the standard deviation of the return distribution. Being pessimistic about the distributional state-value function leads to more stable learning, reduces the variance, and, tends to improve the robustness of systems as demonstrate (Brekelmans et al., 2022). Recent advances in Robust MDPs have shown a link between this field and Regularised MDPs as in Derman et al. (2021); Kumar et al. (2022).

**Distributional RL :** Second-order estimation is done using Distributional Reinforcement Learning (Bellemare et al., 2017; Zhang & Weng) using a quantile estimate of our distribution to approximate our action value function (Dabney et al., 2017; 2018) with the QRDQN and IQN algorithms. Distributional state-action function representation is also used to learn an accurate critic for a policy-based algorithm, such as in Kuznetsov et al. (2020); Ma et al. (2021); Nam et al. (2021).

**Risk-Averse RL :** Risk-averse RL aims at minimizing different objectives than the classical mean optimization e.g. CVaR or other risk measures. For example, Dabney et al. (2018); Ma et al. (2021) use distributional RL for optimizing different risk measures. Our goal is to show the robustness of using risk-averse solutions to our initial problem. Our formulation is close to mean-variance formulation (Jain et al., 2021b; Wang & Zhou, 2020) that already exists in risk-averse RL, although not using a distributional framework that shows highly competitive performance in a controlled setting.

**Pessimism and Optimism in Distributional RL** Moskovitz et al. (2021) describes a way of performing Optimistic / Pessimistic Deep RL using a constructed confidence interval with the variance of rewards. Their work is close to ours in the pessimistic case but the confidence interval is expressed in terms of variance of expectation estimate and not using the variance of the distribution itself. Moreover, they use an adaptive regularizer where we look at the interest of using a fixed parameter.

**Preliminaries:** Taking into account a Markov Decision Process (MDP) $(\mathcal{S}, \mathcal{A}, P, \gamma)$, where $\mathcal{S}$ is the state space, $\mathcal{A}$ is the action space, $P(r, s' \mid s, a)$ is the reward and transition distribution from state $s$ to $s'$ taking action $a$ and $\gamma \in (0, 1)$ is the discount factor. Stochastic policies are denoted $\pi(a \mid s) : \mathcal{S} \to \Delta(\mathcal{A})$ and we consider the cases of action space either discrete our continuous. A rollout or trajectory using $\pi$ from state $s$ using initial action $a$ is defined as the random sequence $\tau^{P,\pi}(s, a) = ((s_0, a_0, r_0), (s_1, a_1, r_1), \ldots)$ with $s_0 = s, a_0 = a, a_t \sim \pi(\cdot \mid s_t)$ and $(r_t, s_{t+1}) \sim P(\cdot, \cdot \mid s_t, a_t)$; we denote the distribution on rollouts by $\mathbb{P}(\tau)$ with $\mathbb{P}(\tau) =$

$P_0(s_0) \prod_{t=0}^{T} P(s_{t+1}, r_t \mid s_t, a_t) \pi(a_t \mid s_t) d\tau$ and generally write $\tau \sim \mathbb{P} = (P, \pi)$. Moreover, we consider the distribution of discounted cumulative return $Z^{P,\pi}(s, a)$ such that $R(\tau^{P,\pi}(s, a)) = \sum_{t=0}^{\infty} \gamma^t r_t \sim Z^{P,\pi}(s, a)$. Finally, the $Q$-function $Q^{P,\pi} : \mathcal{S} \times \mathcal{A} \to \mathbb{R}$ of $\pi$ is the expected discounted cumulative return of the distribution, defined as follows.

$$Q^{P,\pi}(s, a) := \mathbb{E}[Z^{P,\pi}(s, a)] = \mathbb{E}_{P,\pi}[R(\tau) \mid a_t \sim \pi(\cdot \mid s_t), (r_t, s_{t+1}) \sim P(\cdot, \cdot \mid s_t, a_t), s_0 = s, a_0 = a].$$

The classical initial goal of RL, also called risk neutral RL, is to find the optimal policy $\pi^*$ where $Q^{P,\pi^*}(s, a) \geq Q^{P,\pi}(s, a)$ for all $\pi$ and $s \in \mathcal{S}, a \in \mathcal{A}$. Finally, the Bellman operator $\mathcal{T}^{\pi}$ and the Bellman optimal operator $\mathcal{T}^*$ are defined as follows:

$$\mathcal{T}^{\pi} Q(s, a) := r(s, a) + \gamma \mathbb{E}_{P,\pi}[Q(s', a')]$$
$$\mathcal{T}^* Q(s, a) := r(s, a) + \gamma \mathbb{E}_P\left[\max_{a'} Q(s', a')\right].$$

Applying either operator from an initial $Q^0$ converges to a fixed point $Q^{\pi}$ or $Q^*$ at a geometric rate as both operators are contractive. Simplifying the notation with regard to $s, a, \pi$ and $P$, we define the set of greedy policies w.r.t. $Q$ called $\mathcal{G}(Q) = \arg\max_{\pi \in \Pi} \langle Q, \pi \rangle$. A classical approach to estimating an optimal policy is known as Approximate Modified Policy Iteration (AMPI) Scherrer et al. (2015)

$$\begin{cases} \pi_{k+1} \in \mathcal{G}(Q_k) \\ Q_{k+1} = (T^{\pi_{k+1}})^m Q_k + \epsilon_{k+1} \end{cases},$$

which usually reduces to Approximate Value Iteration (AVI, $m = 1$) and Approximate Policy Iteration (API, $m = \infty$) as special cases. The term $\epsilon_{k+1}$ accounts for errors made when applying the Bellman operator in RL algorithms with stochastic approximation.

## 2 ROBUST FORMULATION USING $\chi^2$-DIVERGENCE CONSTRAINTS.

In this section, we would like to find a policy that is robust to a change of environment law $P$, as small variations of $P$ should not have a big impact on the new policy in the greedy step. In our case we are not looking at the classical greedy step $\pi' \in \mathcal{G}(Q) = \arg\max_{\pi \in \Pi} \langle Q, \pi \rangle$, but rather at the following greedy step :

$$\pi' \in \mathcal{G}(Q) = \arg\max_{\pi \in \Pi} \langle \min_P Q^{(P,\pi)}, \pi \rangle.$$

This heuristic in the greedy step can also be interpreted at trying to avoid an overestimation of the Q functions present in the Deep RL algorithms. Using this formulation, we need to constrain the set of admissible transitions from the state action to the next state $P$ to get a solution to the problem. In general, without constraint, the problem is NP-Hard, it therefore requires constraining the problem to specific distributions that are not too far from the original one using a distance between distributions such as the Wasserstein metric (Abdullah et al., 2019) or other specific distances if the problem can be simplified (Eysenbach & Levine, 2021). Furthermore, if an explicit form of $\min_P Q^{(P,\pi)}$ could be calculated exactly for a given divergence, it would lead to a simplification of this max-min optimization problem into a simple maximisation one. In fact, simplification of the problem is possible using a specific $f$-divergence denoted $\mathcal{H}_f$ to constrain the problem with $f$ a close convex function such that $f : \mathbb{R} \to \mathbb{R} \cup \{+\infty\}$ and $f(z) \geq f(1) = 0$ for all $z \in \mathcal{R}$ :

$$\mathcal{H}_f(\mathbb{Q} \mid \mathbb{P}) = \left\{ \begin{array}{ll} \sum_{i:p_i>0} p_i f\left(\frac{q_i}{p_i}\right) & ; \quad \sum_{i:p_i>0} q_i = 1, q_i \geq 0 \\ +\infty \quad \text{otherwise.} \end{array} \right\}.$$

with $\mathbb{P}, \mathbb{Q}$ two probability measures. This constraint requires $q_i = 0$ if $p_i = 0$ which makes the measure $\mathbb{Q}$ absolutely continuous with respect to $\mathbb{P}$. The $\chi^2$-divergence is a particular case of $f$-divergence with $f(z) = (z-1)^2$. For trajectories sampled from the distribution $P_0$ and looking at distribution $P$ close to $P_0$ with regard to the $\chi^2$-divergence, the minimisation problem reduces to :

$$\min_{P \in D_{\chi^2}(P\|P_0) \leq \alpha} Q^{(P,\pi)} = Q^{(P_0,\pi)} - \alpha \mathbb{V}[Z^{P_0}]^{\frac{1}{2}} = \mathbb{E}[Z^{P_0,\pi}] - \alpha \mathbb{V}[Z^{P_0,\pi}]^{\frac{1}{2}}. \tag{1}$$

The proof can be found in Appendix A for $\alpha$ such that $\alpha \leq \frac{\mathbb{V}[Z^{P_0}]}{\|\tilde{Z}^{P_0}\|_\infty^2} \leq 1$ with $\tilde{Z}^{P_0} = Z^{P_0} - \mathbb{E}[Z^{P_0}]$ the centered return distribution and $\mathbb{V}[Z^{P_0}]$ the variance of returns. For $\alpha > \frac{\mathbb{V}[Z^{P_0}]}{\|\tilde{Z}^{P_0}\|_\infty^2}$, the equality becomes an inequality, but we still optimize a lower bound of our initial problem. Defining a new greedy step which is penalized by the standard deviation :

$$\pi' \in \mathcal{G}_\alpha(Q) = \arg\max_{\pi \in \Pi} \langle \min_{\mathcal{P} \in D_{\chi^2}(P\|P_0) \leq \alpha} Q^{(\mathcal{P},\pi)}, \pi \rangle = \arg\max_{\pi \in \Pi} \langle Q^{(P_0,\pi)} - \alpha \mathbb{V}[Z^{(P_0,\pi)}]^{\frac{1}{2}}, \pi \rangle,$$

we now consider the following scheme :

$$\begin{cases} \pi_{k+1} \in \mathcal{G}_\alpha(Q_k) \\ Q_{k+1} = (T\pi_{k+1})^m Q_k + \epsilon_{k+1} \end{cases}. \tag{2}$$

Approximation identities such as (1) for a larger class of $f$-divergences and not only $\chi^2$ can be found in the work of (Duchi et al., 2016). This work motivates the use of the $f$-divergence as a measure of dissimilarity to improve robustness in our problem, since robust mean estimation and mean-standard deviation optimisation are closed. In the recent work of Kumar et al. (2022) using the $L_p$ norm for constraints and the tabular case, similar results are derived, and for the particular case of the $L_2$, a standard deviation term also appears as a regularization similarly to our work.

This idea is also very close to Risk-averse formulation in RL (i.e minimizing risk measure and not only the mean of rewards) as a mean standard deviation objective is used but here the idea is to approximate a robustness problem in RL. To do so, the standard deviation of the distribution of the returns must be estimated. Many ways are possible, but we favour distributional RL using a distributional representation of the critic (Bellemare et al., 2017; Dabney et al., 2017; 2018) which achieve very good performances in many RL applications. Estimating the quantiles of the distribution of return, we can simply estimate the standard deviation using the classical estimator of the standard deviation given the quantiles over an uniform grid $\{q_i(s,a)\}_{1 \leq i \leq n}, \forall(s,a) \in \mathcal{S} \times \mathcal{A}$ :

$$\mathbb{V}[Z(s,a)]^{\frac{1}{2}} = \sigma(s,a) = \Big( \sum_{i=1}^{n} (q_i(s,a) - \bar{q}(s,a))^2 \Big)^{\frac{1}{2}}$$

where $\bar{q}$ is the classical estimator of the mean. A different interpretation of this formulation could be that by taking actions with less variance, we construct a confidence interval with the standard deviation of the distribution

$$Z^\pi(s,a) \stackrel{d}{=} \bar{Z}(s,a) - \alpha\sigma(s,a),$$

where $\bar{Z}(s,a)$ is the mean of the distribution. This idea is present in classical UCB algorithms (Auer, 2002) or pessimism/optimism Deep RL. Here, we construct a confidence interval using the distribution of the return and not different estimates of the $Q$ function such as in Moskovitz et al. (2021); Bai et al. (2022). In the next section, we derive two algorithms, one for a discrete action space and one for a continuous action space, using this idea. A very interesting way of doing robust learning is by doing Max entropy RL such as in the SAC algorithm. In Eysenbach & Levine (2021), a demonstration that SAC is a surrogate of Robust RL is demonstrated formally and numerically and we will compare our algorithm to this method.

## 3 ALGORITHMS BASED ON DISTRIBUTIONAL RL

To derive our algorithms, an estimation of the second-order moment of the distribution of return must be carried out. For discrete action space a variant of QR-DQN (Dabney et al., 2017) with a mean-standard deviation objective is proposed whereas, for continuous action space, we propose a mean-standard TQC algorithm (Kuznetsov et al., 2020) based on the soft-actor framework as it has already shown some robustness as a surrogate of Robust RL (Eysenbach & Levine, 2021).

### 3.1 DISTRIBUTIONAL RL USING A QUANTILE REPRESENTATION

Distributional RL aims at approximating the distribution of returns $Z^\pi(s,a)$ rather than $Q^\pi(s,a) := \mathbb{E}[Z^\pi(s,a)]$ as in the classical RL framework. Many algorithms and distributional representation

of the critic exist (Bellemare et al., 2017; Dabney et al., 2017; 2018) but here we focus on QR-DQN Dabney et al. (2017) that approximates the distribution of returns $Z^\pi(s, a)$ with $Z_\psi(s, a) := \frac{1}{M} \sum_{m=1}^{M} \delta\left(\theta_\psi^m(s, a)\right)$, a mixture of atoms-Dirac delta functions located at $\theta_\psi^1(s, a), \ldots, \theta_\psi^M(s, a)$ given by a parametric model $\theta_\psi : \mathcal{S} \times \mathcal{A} \to \mathbb{R}^M$. Parameters $\psi$ of a neural network are obtained by minimizing the average over the 1-Wasserstein distance between $Z_\psi$ and the temporal difference target distribution $\mathcal{T}_\pi Z_{\bar\psi}$, where $\mathcal{T}_\pi$ is the distributional Bellman operator defined in Bellemare et al. (2017). The control version or optimal operator is denoted $\mathcal{T} Z_{\bar\psi}$,

$$\mathcal{T}^\pi Z(s, a) = \mathcal{R}(s, a) + \gamma Z(s', a') \text{ with } s' \sim \mathcal{P}(\cdot \mid s, a), a' \sim \pi(\cdot \mid s').$$

Considering $\mathcal{Z}$ being the space of action-value distributions with finite moments: $\mathcal{Z} = \{Z : \mathcal{X} \times \mathcal{A} \to P(\mathbb{R})\}$ with $\mathbb{E}\left[|Z(x, a)|^p\right] < \infty, \forall(x, a), p \geq 1$ Bellemare et al. (2017) show that :

$$\|\mathbb{E}\mathcal{T} Z_1 - \mathbb{E}\mathcal{T} Z_2\|_\infty \leq \gamma \|\mathbb{E}Z_1 - \mathbb{E}Z_2\|_\infty$$

. This proves that point wise convergence is exponentially fast for the mean of the distribution as in the classical RL problem. According to Dabney et al. (2017), the minimization of the 1-Wasserstein loss can be done by learning quantile locations for fractions $\tau_m = \frac{2m-1}{2M}, m \in [1..M]$ via quantile regression loss, defined for a quantile fraction $\tau \in [0, 1]$ as :

$$\mathcal{L}_{\text{QR}}^\tau(\theta) := \mathbb{E}_{\tilde{Z} \sim Z}\left[\rho_\tau(\tilde{Z} - \theta)\right], \text{ with}$$
$$\rho_\tau(u) = u(\tau - \mathbb{I}(u < 0)), \forall u \in \mathbb{R}.$$

Finally, to obtain better gradients when $u$ is small, the (asymmetric) Huber quantile loss is used:

$$\rho_\tau^H(u) = |\tau - \mathbb{I}(u < 0)|\mathcal{L}_H^1(u),$$

where $\mathcal{L}_H^1(u)$ is a classical Huber loss with parameter 1. The quantile representation has the advantage of not fixing the support of the learned distribution, and is used to represent the distribution of return in our algorithm for both discrete and continuous action space.

## 3.2 MEAN-STANDARD DEVIATION RL WITH DISCRETE ACTION SPACE

Once the state action return distribution is estimated, a phase of policy improvement is performed using a Q-learning style algorithm with distributional estimation such as QR-DQN (Dabney et al., 2017). The main difference in our case is that we do not take the expectation in this phase, but the mean standard deviation objective (3) in the greedy step using $M$ quantiles over a uniform grid in $[0, 1]$. Formally, we choose actions with less variance to improve robustness using a mean standard deviation objective, where the classical empirical estimator of the standard deviation using quantiles of the distribution is used. The estimation step of the algorithm does not change as we use the classical Bellman operator as in the classical QR-DQN algorithm. The parameters $\psi$ of the quantile network are classically updated using a stochastic gradient descent, where $\hat\nabla$ represents a stochastic estimate of the gradient. Moreover, $\beta$ is the parameter of soft or Polyak's update of the target quantile network parametrized by $\bar\psi$ in algorithm 1. Finally, $\mathcal{D}$ represents the replay buffer where we store all different transitions $(s, a, s', r)$ and $\xi : Z \to \mathbb{E}[Z] - \alpha\mathbb{V}[Z]^{\frac{1}{2}}$ is the objective function that we optimize.

$$a^* = \arg\max_{a \in \mathcal{A}} \xi_\alpha Z^\pi(s, a) = \arg\max_{a \in \mathcal{A}} \mathbb{E}[Z^\pi(s, a)] - \alpha\sqrt{\mathbb{V}[Z^\pi(s, a)]} \quad (3)$$

## 3.3 MEAN-STANDARD DEVIATION MAXIMUM ENTROPY RL FOR CONTINUOUS ACTION SPACE

We use a distributional maximum entropy framework for the continuous action space that is close to the TQC algorithm Kuznetsov et al. (2020). This method uses an actor-critic framework with a distributional truncated critic to ovoid overestimation in the estimation with the max operator. This algorithm is based on a soft-policy iteration, where we penalize the target $y_i(s, a)$ using the entropy of the distribution. More formally, to compute the target, the principle is to train $N$ approximate estimate $Z_{\psi_1}, \ldots Z_{\psi_C}$ of the distribution of returns $Z^\pi$ where $Z_{\psi_c}$ maps each $(s, a)$ to $Z_{\psi_c}(s, a) := \frac{1}{M} \sum_{m=1}^{M} \delta\left(\theta_{\psi_n}^m(s, a)\right)$, which is supported on atoms $\theta_{\psi_c}^1(s, a), \ldots, \theta_{\psi_c}^M(s, a)$. Then approximations $Z_{\psi_1}, \ldots Z_{\psi_N}$ are trained on the temporal difference target distribution denoted

---

**Algorithm 1** QR-DQN with Standard Deviation penalisation

> **Initial** critics $Z_\psi, Z_{\bar\psi}$
> **for** each iteration **do**
>    **for** each step of the environment **do**
>       collect $(s_t, a_t, r_t, s_{t+1})$ according to $\pi(a_t|s_t) = \arg\max_a \xi_\alpha Z^\pi(s_t, a_t)$
>       $\mathcal{D} \leftarrow \mathcal{D} \cup \{(s_t, a_t, r_t, s_{t+1})\}$
>    **end for**
>    **for** each gradient steps **do**
>       Sample batch $(s, a, r, s')$ of $\mathcal{D}$
>       Take $a^* = \arg\max_{a'} \xi_\alpha Z^\pi(s', a')$
>       $y_i(s, a) \leftarrow r + \gamma\theta^i_\psi(s', a^*), i \in [1..M]$
>       $J_Z(\psi) = \mathbb{E}_{\mathcal{D},\pi} \sum_{i,j=1}^m \rho^H_{\tau_j}\left(y_i(s, a) - \theta^j_\psi(s, a)\right)$
>       $\psi \leftarrow \psi - \lambda_Z \hat\nabla_\psi J_Z(\psi),$
>       $\bar\psi \leftarrow (1 - \beta)\bar\psi + \beta\psi$
>    **end for**
> **end for**
> **return** critic $Z_\psi, Z_{\bar\psi}$.

$Y(s, a)$ constructed as follows. First atoms of trained distributions $Z_{\psi_1}(s', a'), \ldots, Z_{\psi_C}(s', a')$ are pooled into $\mathcal{Z}(s', a') := \left\{ \theta^m_{\psi_c}(s', a') \mid c \in [1..C], m \in [1..M] \right\}$. We denote elements of $\mathcal{Z}(s', a')$ sorted in ascending order by $z_{(i)}(s', a')$, with $i \in [1..MC]$. Then we only keep the $kC$ smallest elements of $\mathcal{Z}(s', a')$. We remove outliers of distribution to avoid overestimation of the value function. Finally, the atoms of the target distribution $Y(s, a) := \frac{1}{kC}\sum_{i=1}^{kC} \delta(y_i(s, a))$ are computed according to a soft policy gradient method where we penalised with the log of the policy :

$$y_i(s, a) := r(s, a) + \gamma\left[z_{(i)}(s', a') - \eta\log\pi_\phi(a' \mid s')\right]. \tag{4}$$

The entropic term $\eta\log\pi_\phi(a' \mid s')$ is also added like in classical SAC algorithm. It encourages exploration and usually improve speed of convergence. As in QR-DQN, the 1-Wasserstein distance between each of $Z_{\psi_n}(s, a), n \in [1..N]$ and the temporal difference target distribution $Y(s, a)$ is minimized learning the locations for quantile fractions $\tau_m = \frac{2m-1}{2M}, m \in [1..M]$. Similarly, we minimize the loss:

$$J_Z(\psi_c) = \mathbb{E}_{\mathcal{D},\pi}\left[\mathcal{L}^k(s_t, a_t; \psi_c)\right] = \mathbb{E}_{\mathcal{D},\pi}\left[\frac{1}{MkC}\sum_{j=1}^M\sum_{i=1}^{kC}\rho^H_{\tau_j}\left(y_i(s, a) - \theta^j_{\psi_c}(s, a)\right)\right] \tag{5}$$

over the parameters $\psi_n$, for each critic. With this formulation, the learning of all quantiles $\theta^m_{\psi_n}(s, a)$ is dependent on all atoms of the truncated mixture of target distributions. To optimize the actor, the following loss based on KL-divergence denoted $D_{KL}$ is used for soft policy improvement :

$$J_{\pi,\alpha}(\phi) = \mathbb{E}_{\mathcal{D}}\left[D_{KL}\left(\pi_\phi(\cdot \mid s) \,\middle\|\, \frac{\exp\left(\frac{1}{\eta}\xi_\alpha(\theta_\psi(s, \cdot))\right)}{D}\right)\right]$$

where $\eta$ can be seen as a temperature and needs to be tuned and D is a constant of normalisation. This expression simplify into :

$$J_{\pi,\alpha}(\phi) = \mathbb{E}_{\mathcal{D},\pi}\left[\eta\log\pi_\phi(a \mid s) - \frac{1}{C}\sum_{c=1}^C \xi_\alpha(\theta_{\psi_c}(s, a))\right] \tag{6}$$

where $s \sim \mathcal{D}, a \sim \pi_\phi(\cdot \mid s)$. Non-truncated estimate of the Q-value are used for policy optimization to avoid a double truncation, in fact the $Z$-functions already approximate truncated future distribution. Finally, $\eta$ is the entropy temperature coefficient and is dynamically adjusted by taking a gradient step with respect to the loss like in Haarnoja et al. (2018) :

$$J(\eta) = \mathbb{E}_{\mathcal{D},\pi_\phi}\left[(-\log\pi_\phi(a_t \mid s_t) - \mathcal{H}_\eta)\eta\right]$$

---

**Algorithm 2** TQC with Standard Deviation penalisation

---

**Initialize** policy $\pi_\phi$, critics $Z_{\psi_c}, Z_{\bar{\psi}_c}$ for $c \in [1..C]$
**for** each iteration **do**
    **for** each step of the environment **do**
        collect $(s_t, a_t, r_t, s_{t+1})$ with policy $\pi_\phi$
        $\mathcal{D} \leftarrow \mathcal{D} \cup \{(s_t, a_t, r_t, s_{t+1})\}$
    **end for**
    **for** each gradient steps **do**
        Sample batch $(s, a, s', r)$ of $\mathcal{D}$
        $y_i(s, a) \leftarrow r(s, a) + \gamma \left[ z_{(i)}(s', a') - \eta \log \pi_\phi(a' \mid s') \right]$
        $\eta \leftarrow \eta - \lambda_\eta \hat{\nabla}_\eta J(\eta)$
        $\phi \leftarrow \phi - \lambda_\pi \hat{\nabla}_\phi J_{\pi,\alpha}(\phi)$
        $\psi_c \leftarrow \psi_c - \lambda_Z \hat{\nabla}_{\psi_c} J_Z(\psi_n), c \in [1..C]$
        $\bar{\psi}_c \leftarrow \beta \psi_c + (1-\beta)\bar{\psi}_c, c \in [1..C]$
    **end for**
**end for**
**return** $\pi_\phi$, critics $Z_{\psi_c}, c \in [1..C]$.

---

at every time the $\pi_\phi$ changes. The target entropy $\mathcal{H}_\eta$ usually is set heuristically $-\mathrm{Dim}(\mathcal{A})$. Temperature $\eta$ decreases if the policy entropy, $-\log \pi_\phi(a_t \mid s_t)$, is higher than $\mathcal{H}_\eta$ and increases otherwise. The algorithm is summarized in Algorithm 2:

Our algorithm is based on SAC but with distributional critics to improve the estimation of $Q$ functions while using the mean standard deviation objective in policy to improve robustness.

## 4 EXPERIMENTS

We try different experiments on continuous and discrete action space to demonstrate the interest of our algorithms for robustness using $\xi : Z \to \mathbb{E}[Z] - \alpha \mathbb{V}[Z]^{\frac{1}{2}}$ instead of the mean. The choice of $\alpha$ is crucial as it determines the degree of penalty in the objective. The more the environment is penalized, the more a pessimistic action is chosen.

### 4.1 RESULTS ON CONTINUOUS ACTION SPACES

For continuous action space, we compare our algorithm with SAC which achieves state-of-the-art robust control (Eysenbach & Levine, 2021) on the Mujoco environment such as Hopper-v3, Walker-v3 or HalfCheetah-v3. We use a version where the entropy coefficient is adjusted during learning for both SAC and our algorithm, as it requires less parameter tuning. Moreover, we show the influence of a distributional critic without a mean-standard deviation greedy step using $\alpha = 0$ to demonstrate the advantage of using a distributional critic against the classical SAC algorithm. We also compare our results to TQC algorithm, varying the penalty $\alpha$ to show that for the tested environment, there exists a value of $\alpha$ such that prediction are more robust to change of dynamics.

The interest of our algorithm is best shown in stochastic environments, since it involves the distributions of returns which are varying in stochastic environments. The only source of stochasticity in the Mujoco subject is the initial point, so in order to make its environments stochastic we have noised environments at each step by adding a noise in $[-1e^{-2}, 1e^{-2}]$ to each action. Since we also compare our algorithm in non-stochastic environments, we differentiate the two cases by denoting noisy environments by **(N)** and environments without noise **(wN)**. In these simulations, variations of dynamics are carried out by moving the relative mass, which is an influential physical parameter in all environments. All algorithms are trained with a relative mass of 1 and then tested on new environment where the mass varies from $0.5$ to $2$. Two phenomena can be observed for the 3 environments.

First, for all environments in Fig 1,2, and Fig 7 in annex, where performance is normalized by the maximum of the performance for every curve to highlight robustness and not only mean-performance. We see that we can find a value of $\alpha$ where the robustness is clearly improved without deteriorating the average performance. In fact, if a penalty is applied too strongly, the average performance can

be reduced, as in the HalfCheetah-v3 environment. For Hopper-v3, a $\alpha$ calibrated at 5 gives very good robustness performances, while for Walker2d-v3, the value is closer to 2. This phenomenon was expected and was in agreement with our formulation. Moreover, our algorithm outperforms the SAC algorithm for Robustness tasks in all environments. Tuning of $\alpha$ must be chosen carefully, for example, $\alpha$ is chosen in $\{0, 1, ..., 5\}$ for Hopper-v3 and Walker2d-v3 whereas values of $\alpha$ are chosen smaller in $\{0, 0.1, 0.5.1, 1.5, 2\}$ and not in a bigger interval. As a rule of thumb for choosing $\alpha$, we can look at the empirical mean and variance at the end of the trajectories to see if the environment has rewards that fluctuate a lot. The smaller the mean/variance ratio, the more likely we are to penalise our environment. For HalfCheetah, the mean/variance ratio is about approximately 100, so we will favour smaller penalties than for Walker2d where the mean/variance ratio is about 50 or 10 for Hopper.

The second surprising observation is that penalizing our objective also improves performance in terms of stability during training and in terms of average performance, especially for Hopper and Walker2d in Fig 4 or sometimes in Fig 3. Similar results are present in the work of (Moskovitz et al., 2021), which gives an interpretation in terms of optimism and pessimism for environments. This phenomenon is not yet explained, but it is present in environments that are particularly unstable and have a lot of variance. The variance of the return is a consequence of the stochasticity of the environment or of the policy. Intuitively, the most favorable settings are thus the one with the most stochasticity. We have, however, observed that our method remains interesting in low-stochasticity or non-stochasticity environments even if the policy is not stochastic. A possible explanation is a better exploration thanks to the pessimistic approach.

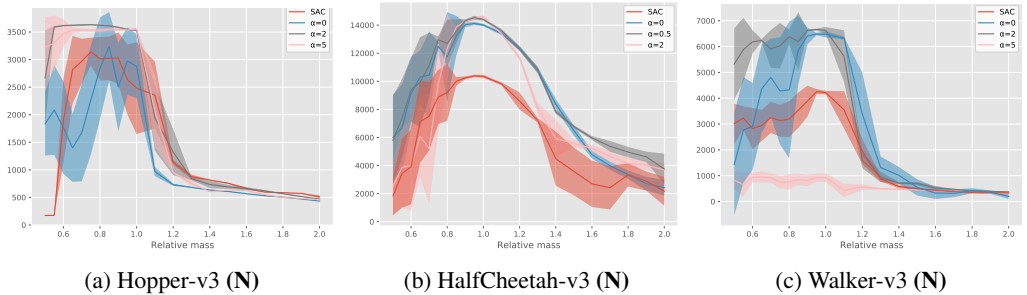

(a) Hopper-v3 **(N)**      (b) HalfCheetah-v3 **(N)**      (c) Walker-v3 **(N)**

Figure 1: y-axis : normalised mean $\pm$ standard deviation over 20 trajectories. x-axis : relative mass.

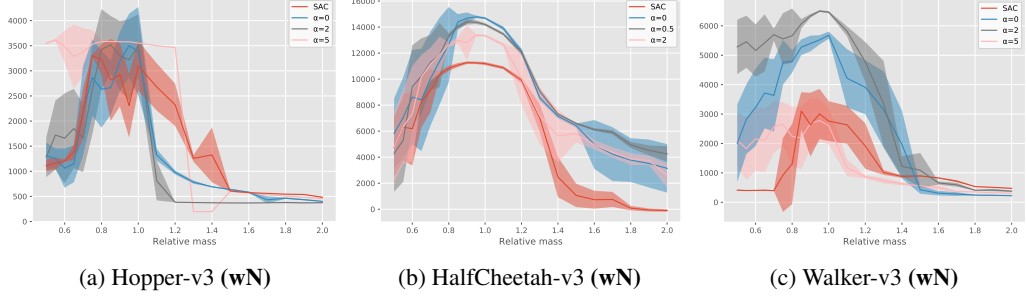

(a) Hopper-v3 **(wN)**      (b) HalfCheetah-v3 **(wN)**      (c) Walker-v3 **(wN)**

Figure 2: y-axis : mean $\pm$ standard deviation over 20 test trajectories. x-axis: relative mass.

## 4.2 RESULTS ON DISCRETE ACTION SPACES

We test our QRDQN algorithm with standard deviation penalization on discrete action space, varying the length of the pole in Cartpole-v1 and Acrobot-v1 environments. We observe similar results for the discrete environment in terms of robustness. Training is done for a length of the pole equal to the x-axis of the black star on the graph, and then for testing, the length of the pole is increased or decreased. We show that robustness is increased when we penalised our distributional critic. We have compared our algorithm to PPO which has shown relatively good results in terms of robustness for discrete action space in (Abdullah et al., 2019) as SAC does not apply to discrete action space. The

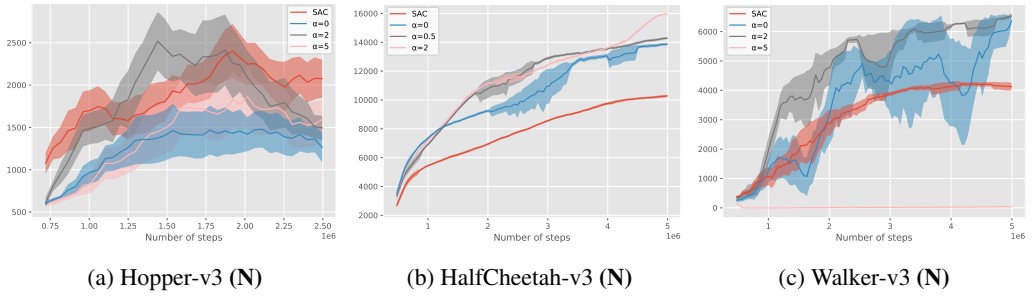

(a) Hopper-v3 (**N**)   (b) HalfCheetah-v3 (**N**)   (c) Walker-v3 (**N**)

Figure 3: y-axis : mean over 20 trajectories ± standard deviation in function of timesteps.

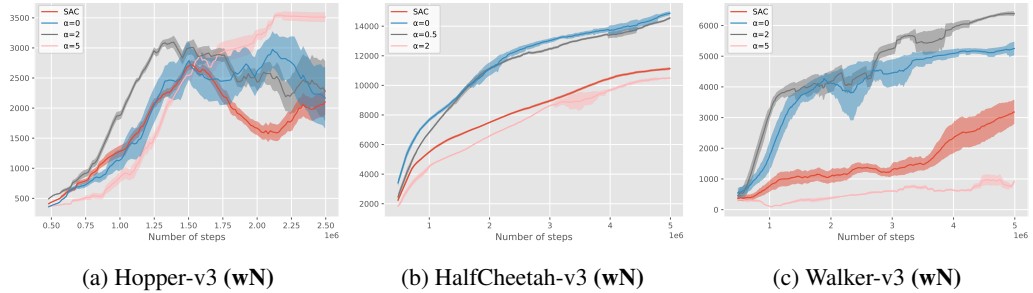

(a) Hopper-v3 (**wN**)   (b) HalfCheetah-v3 (**wN**)   (c) Walker-v3 (**wN**)

Figure 4: y-axis : mean over 20 trajectories ± standard deviation in function of timesteps.

same phenomenon is observed in terms of robustness as for continuous environments. However, the improvement in terms of mean performance on Hopper and Walker2d environments is not observed. This is partly explained by the fact that the maximum reward is reached in Cartpole and Acrobot quickly. An ablation study can be found in annex C where we study the impact of penalization on our behavior policy during testing and on the policy used during learning. It is shown that both are needed in the algorithm.

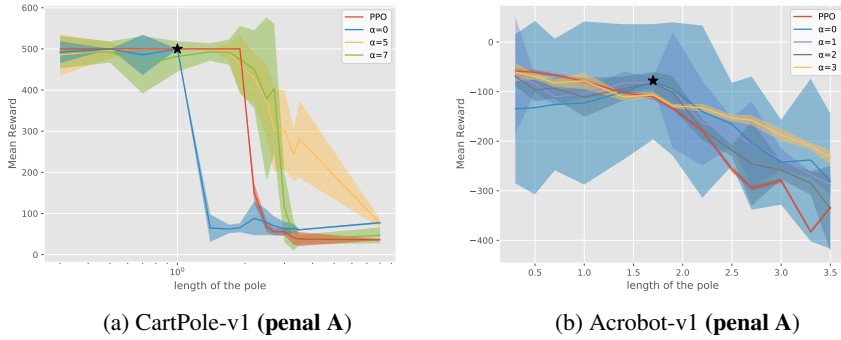

(a) CartPole-v1 (**penal A**)   (b) Acrobot-v1 (**penal A**)

Figure 5: Mean over 20 trajectories varying length's pole, trained on the x-axis of the black star.

## 5 CONCLUSION

In this paper, we show that by using a mean-standard deviation formulation to choose our actions pessimistically, we can increase the robustness of our environment for continuous and discrete environments. A single fixed $\alpha$ parameter must be tuned to obtain good performance without penalizing the average performance too much. Moreover, for some environments, it is relevant to penalize actor to increase the average performance as well when there is a lot of variability in the environment. Theoretical links with a (Kumar et al., 2022) of $L_p$ norms could be an interesting way of comparing our algorithm. The analysis of error propagation in this AMPI scheme is left for future work to understand theoretically how our algorithm behaves.

## 6 ACKNOWLEDGEMENT

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

# A  PROOF OF MEAN-STANDARD DEVIATION FORMULATION AS A ROBUST PROBLEM

We consider the following equality :

$$\min_{P \in D_{\chi^2}(P\|P_0) \le \alpha} Q^{(P,\pi)} = Q^{(P_0,\pi)} - \alpha \mathbb{V}[Z^{P_0,\pi}]^{\frac{1}{2}}. \tag{7}$$

Consider that trajectories $\tau$ is drawn from $\mathbb{P}$ but here we will write $P$ the transition of the environment as the policy $\pi$ is fixed and it is the only part which differ.

Writing $\tilde{R}(\tau) = R(\tau) - \mathbb{E}_{\tau \sim P_0}[R(\tau)]$ we get :

$$
\begin{aligned}
\|\mathbb{E}_{\tau \sim P}[R(\tau)] - \mathbb{E}_{\tau \sim P_0}[R(\tau)]\| &= \left\| \int_\tau \tilde{R}(\tau)\big(p(\tau) - p_0(\tau)\big)d\tau \right\| \\
&= \left\| \int_\tau \tilde{R}(\tau)\sqrt{p_0(\tau)}\frac{\big(p(\tau) - p_0(\tau)\big)}{\sqrt{p_0(\tau)}}d\tau \right\| \\
&\le \left\| \int_\tau \tilde{R}(\tau)^2 p_0(\tau)d\tau \right\|^{\frac{1}{2}} \left\| \int_\tau \frac{\big(p(\tau) - p_0(\tau)\big)^2}{p_0(\tau)}d\tau \right\|^{\frac{1}{2}} \\
&= \mathbb{V}_{P_0}[R(\tau)]^{\frac{1}{2}} D_{\chi^2}(P\|P_0)^{\frac{1}{2}},
\end{aligned}
$$

because of the positivity of the divergence and of the variance, the norms are removed. This inequality comes from the Cauchy-Schwarz inequality and becomes an equality if for $\lambda \in \mathbb{R}$ :

$$\tilde{R}(\tau)p_0(\tau) = \lambda(p(\tau) - p_0(\tau)) \iff p(\tau) = p_0(\tau)(1 + \frac{1}{\lambda}\tilde{R}(\tau)). \tag{8}$$

However, $p(\tau)$ needs to be non-negative and sum to one as it is a measure. However, the normalization condition is respected by construction, to ensure that the measure is non-negative, this requires $\left\| \tilde{R}(\tau)/\lambda \right\| \le 1$ in the case where $\lambda \le 0$ . In this case of equality, we obtain from 8 that $D_{\chi^2}(P\|P_0) = \frac{\mathbb{V}_{P_0}[R(\tau)]}{\lambda^2}$. Replacing the divergence in the inequality, the following result holds :

$$\|E_{\tau \sim P}[R(\tau)] - E_{\tau \sim P_0}[R(\tau)]\| \le \frac{\mathbb{V}_{P_0}(R(\tau))}{\lambda}.$$

To prove (7) we are interested in the case where $D_{\chi^2}(P\|P_0) \le \alpha$, from the initial inequality we obtain :

$$\min_{P \in D_{\chi^2}(P\|P_0) \le \alpha} Q^{(P,\pi)} \ge \min_{P \in D_{\chi^2}(P\|P_0) \le \alpha} Q^{(P_0,\pi)} - D_{\chi^2}(P\|P_0)\mathbb{V}[Z^{P_0,\pi}]^{\frac{1}{2}} = Q^{(P_0,\pi)} - \alpha \mathbb{V}[Z^{(P_0,\pi)}]^{\frac{1}{2}}$$

with the maximum value of $\alpha$ equals to $D_{\chi^2}(P\|P_0) = \frac{\mathbb{V}_{P_0}[R(\tau)]}{\lambda^2} \le \frac{\mathbb{V}_{P_0}[R(\tau)]}{\|\tilde{R}\|_\infty^2} = \frac{\|\tilde{R}\|_2^2}{\|\tilde{R}\|_\infty^2} \le 1$, where the first inequality comes from the conditions $\left\| \tilde{R}(\tau)/\lambda \right\| \le 1$ and the last one comes from that the $L_2$ norm is smaller than $\infty$-norm.

If our problem is contrained, assuming $\alpha \le \frac{\mathbb{V}_{P_0}[R(\tau)]}{\|\tilde{R}\|_\infty^2} \le 1$, we obtain the following results with the maximum attained for $D_{\chi^2}(P\|P_0) = \alpha$ :

$$\min_{\mathcal{P} \in D_{\chi^2}(P\|P_0) \le \alpha} Q^{(\mathcal{P},\pi)} = Q^{(\mathcal{P}_0,\pi)} - \alpha \mathbb{V}[Z^{P_0}]^{\frac{1}{2}}. \tag{9}$$

For $\alpha > 1$, we still optimize a lower bound of the quantity of interest. The formulation of our algorithm becomes the following.

$$
\begin{cases}
\pi_{k+1} \in \mathcal{G}_\alpha\,(Z_k) = \mathcal{G}(\xi_\alpha(Z_k) = \arg\max_{\pi \in \Pi}\langle \mathbb{E}[Z_k] - \alpha\sqrt{\mathbb{V}[Z_k]}, \pi\rangle \\
Z_{k+1} = \big(T_\sigma^{\pi_{k+1}}\big)^m Z_k
\end{cases}
.
$$

# B   FURTHER RESULTS ON CONTINUOUS ACTION SPACE

## B.1   NORMALISED RESULTS

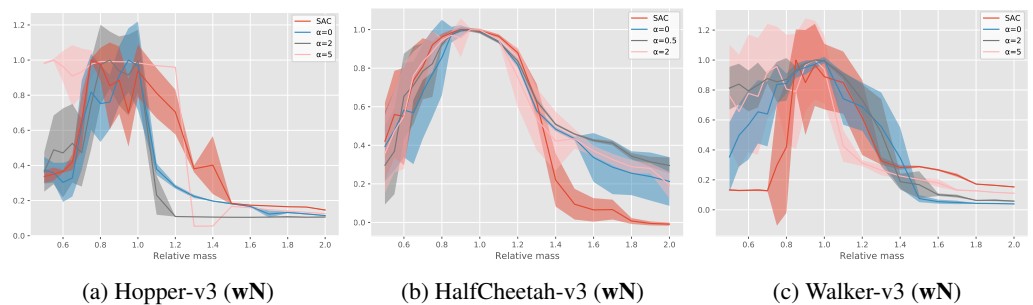

(a) Hopper-v3 (**wN**)                (b) HalfCheetah-v3 (**wN**)                (c) Walker-v3 (**wN**)

Figure 6: y-axis : normalised mean ± standard deviation over 20 trajectories. x-axis : relative mass.

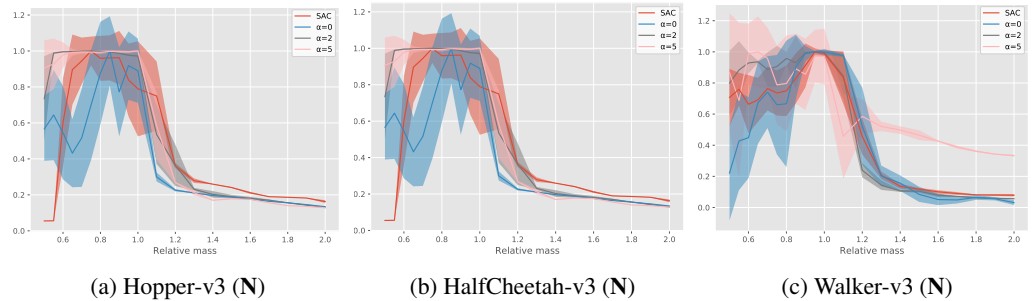

(a) Hopper-v3 (**N**)                (b) HalfCheetah-v3 (**N**)                (c) Walker-v3 (**N**)

Figure 7: y-axis : normalised mean ± standard deviation over 20 trajectories. x-axis : relative mass.

The results were normalised to better reflect the improvement without being biased by the average performance which is higher with a distributional critic.

# C   FURTHER EXPERIMENTAL DETAILS

All experiments were run on a cluster containing an Intel Xeon CPU Gold 6230, 20 cores, and all experiments were performed on a single CPU between 3 and 6 hours for continuous control and less than 1 hour for the discrete control environment.

Pre-trained models will be available for all algorithms and environments on a GitHub link.

The Mujoco OpenAI Gym task licensing information is given at https://github.com/openai/gym/blob/master/LICENSE.md. The baseline implementation of PPO, SAC, TQC, and QRDQN can be found in Raffin et al. (2019). Moreover, hyperparameters across all experiments used are displayed in Table 2, 1 and 3 .

# D   ABLATION STUDY FOR DISCRETE ACTION SPACE ON CARTPOLE-v1

The purpose of this ablation study is to look at the influence of penalization in the discrete action space with QRDQN. In the figures below, we look at the influence of penalizing only during training, which will have the effect of choosing less risky actions during training in order to increase robustness. This curve is denoted *Train penalized.*

Then we look at the influence of penalizing only once the policy has been learned using classic QRDQN without penalization. Only mean-var actions are selected here during testing and not during training. This experience is denoted *Train Penalization*.

Finally, we compare its variants with our algorithm called *Full penalization.* The results of the ablation are: to achieve optimal performance, both phases are necessary.

When penalties are applied only during training. Good performance is generally obtained close to the length 1 where we train our algorithm. However, the performance is difficult to generalize when the pole length is increased,increased, as we do not penalize during testing.

When we penalize only during testing: even if the performances deteriorate, we see that it tends to add robustness because the curves have less tendency to decrease when we increase the length of the pole. The performances are not very high as we play different acts than those taken during the learning.

So both phases are necessary for our algorithm. Penalizing during training allows for safer exploration and penalizing during testing allows for better generalization.

The ablation study for the continuous case is more difficult to do. Indeed, the fact that the penalty occurs only in the gradient descent phase makes it difficult to penalize only in the test phase.

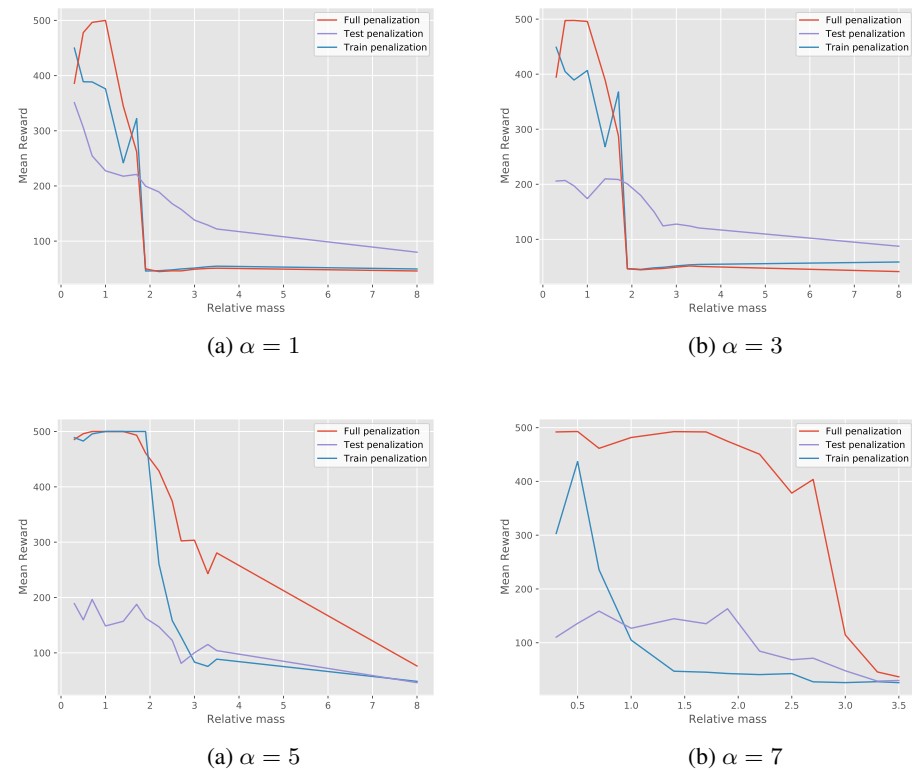

(a) $\alpha = 1$

(b) $\alpha = 3$

(a) $\alpha = 5$

(b) $\alpha = 7$

# E  HYPERPARAMETERS

For HalfCheetah-v3 , penalisation is chosen in $[0, 2]$ and not $[0, 5]$ like in Walker-v3 and Hopper-v3.

Table 1: Table of best hyperparameter for Cartpole-v1

| Hyperparameter | QRDQN with standard deviation penalisation | PPO |
|---|---|---|
| Learning Rate | 2.3e-3 | 3e-4 |
| Optimizer | Adam | Adam |
| Replay Buffer Size | 10e5 | N/A |
| Number of Quantiles | 10 | N/A |
| Huber parameter $\kappa$ | 1 | N/A |
| Penalisation $\alpha$ | {0,1,3,5,7 } | N/A |
| Network Hidden Layers for Policy | N/A | 256:256 |
| Network Hidden Layers for Critic | 256:256 | 256:256 |
| Number of samples per Minibatch | 64 | 256 |
| Discount factor $\gamma$ | 0.99 | 0.99 |
| Target smoothing coefficient $\beta$ | .0.005 | N/A |
| Non-linearity | ReLu | ReLu |
| Target update interval | 10 | N/A |
| Gradient steps per iteration | 1 | 1 |
| Entropy coefficient | N/A | 0 |
| GAE $\lambda$ | 0.95 | 0.8 |

Table 2: Table of best hyperparameter for Acrobot-v1

| Hyperparameter | QRDQN with standard deviation penalisation | PPO |
|---|---|---|
| Learning Rate | 6.3e-4 | 3e-4 |
| Optimizer | Adam | Adam |
| Replay Buffer Size | 50 000 | N/A |
| Number of Quantiles | 25 | N/A |
| Huber parameter $\kappa$ | 1 | N/A |
| Penalisation $\alpha$ | {0, 0.5, 1, 2, 3} | N/A |
| Network Hidden Layers for Critic | 256:256 | 256:256 |
| Network Hidden Layers for Policy | N/A | 256:256 |
| Number of samples per Minibatch | 128 | 64 |
| Discount factor $\gamma$ | 0.99 | 0.99 |
| Target smoothing coefficient $\beta$ | .0.005 | N/A |
| Non-linearity | ReLu | ReLu |
| Target update interval | 250 | N/A |
| Gradient steps per iteration | 4 | 1 |
| Entropy coefficient | N/A | 0 |
| GAE $\lambda$ | 0.95 | 0.95 |

Table 3: Table of best hyperparameter for all continuous environments

| Hyperparameter | TQC with standard deviation penalisation | SAC |
|---|---|---|
| Learning Rate | linear decay from 7.3e-4 | linear decay from 7.3e-4 |
| Optimizer | Adam | Adam |
| Replay Buffer Size | $10^6$ | $10^6$ |
| Expected Entropy Target | $-\dim\mathcal{A}$ | $-\dim\mathcal{A}$ |
| Number of Quantiles | 25 | N/A |
| Huber parameter $\kappa$ | 1 | N/A |
| Penalisation $\alpha$ | $\{0, 1, ...5\}$ | N/A |
| Network Hidden Layers for Policy | 256:256 | 256:256 |
| Network Hidden Layers for Critic | 512:512:512 | 256:256 |
| Number of dropped atoms | 2 | N/A |
| Number of samples per Minibatch | 256 | 256 |
| Discount factor $\gamma$ | 0.99 | 0.99 |
| Target smoothing coefficient $\beta$ | .0.005 | 0.005 |
| Non-linearity | ReLu | ReLu |
| Target update interval | 1 | 1 |
| Gradient steps per iteration | 1 | 1 |

