# OpenReview forum: "Robust Reinforcement Learning with Distributional Risk-averse formulation"
_ICLR.cc/2023/Conference — Submitted to ICLR 2023_

### Official Review · Reviewer_UvPJ · 2022-10-25

**Confidence:** 3
**Correctness:** 3
**Technical Novelty And Significance:** 3
**Empirical Novelty And Significance:** 2
**Recommendation:** 3

**Clarity, Quality, Novelty And Reproducibility:**

-For clarity, some minors:
1. "ovoid" in the last several lines on Page 5
2. The notations of $N or $C$, $n$ and $c$ keep rotating is very confusing in the last several lines on Page 5 and Page 6.
3. The index $i$ in Equation 4 is not mentioned before?
4. Is there a missing of $Z$ in Equation 6 and the equation above Equation 6?
5. There are also some confusing notations in Algorithm 2, such as what is $i$?

-The novelty and originality of this work seem limited. The tailored design of the 'Q-value' does not have convincing results in experimental results.

**Strength And Weaknesses:**

-Strength:
1. The intuition and derivation of the designing algorithm in robust RL are well-explained by showing the relationship between robust RL and mean standard deviation optimization.
2. The presentation and writing of this work are very clear and easy to follow.

-Weaknesses:
1. It seems the main contribution is that this work proposes a different 'Q-value' function by using mean standard deviation optimization and minus $\alpha$ multiplying the standard deviation. However,  the experiments do not show that such designing of 'Q-value' has benefits if I didn't miss something. Such as in Figure 1 and Figure 2, the algorithm with $\alpha=0$ (vanilla Q-value without extra design) and distributed critics works even better or the same as when $\alpha>0$ (proposed methods in this work). It seems there does not exist a  great $\alpha$ for all environments.

**Summary Of The Paper:**

This work targets the robust reinforcement learning problem with an uncertainty set with respect to $\chi^2$. It proposes an alternative 'Q-value' $Z$ and arrives at the mean standard deviation RL. Then the algorithms in the discrete case and continuous case are proposed separately and evaluated in different experiments with comparisons to baselines.

**Summary Of The Review:**

In summary, although this work is motivated by a clear intuition between the robust RL and the mean standard derivation optimization, the proposed technical 'Q-value' does not have convincing results in evaluated experiments. So I suggest for rejection.

---

> ### Author Response · Authors · 2022-11-16
> **Answer to UvPJ**
>
> Thank you for all the feedback and all comments you made.
>
> Some comments on the weaknesses you mentioned :
>
> "However, the experiments do not show that such designing of 'Q-value' has benefits "
>
> The figures in the paper clearly show a gain in robustness and performance for the Hopper and Walker environments in Fig 1 and 2. When the mass parameter is varied, the performances remain more stable over a larger range than without variance penalty. Moreover, most of the time, the mean performance is better. For HalfChettah, the results are less convincing because the environment is more stable. This is why we still think that our tailored design of the 'Q-value'  has convincing results in experimental results.
>
> Thank you again for all your detailed feedback.

---

> ### Comment · Area_Chair_otJQ · 2022-11-21
> **Any comments to the responses from authors?**
>
> Dear Reviewer UvPJ,
>
> Thank you very much for your informative review.  The authors have provided responses to your concerns.  How did they change your evaluation, particularly on the benefit of the approach?

---

### Official Review · Reviewer_anYu · 2022-10-25

**Confidence:** 3
**Correctness:** 3
**Technical Novelty And Significance:** 2
**Empirical Novelty And Significance:** 2
**Recommendation:** 5

**Clarity, Quality, Novelty And Reproducibility:**

The paper is quite clear, although I was not able to check the mathematical derivations

I have doubts about the novelty of the approach, specifically using a mean/std approach as objective function is well established in safe, risk-sensitve, and curiosity based RL and other fields of research (e.g. active learning).


**Strength And Weaknesses:**

Strengths:
- the paper is in a relevant field of research
- the mathematical derivations appear sound
- relevant work is metioned

Weaknesses:

- Novelty:
   - using a mean/std approach for risk-sensitive RL has been around for some time and is standard procedure with many publications
   - the results are very much to be expected, adding a standard deviation makes training more stable at the cost of performance

- Flexibility:   Compared to  Smirnova et al. (2019) the authors state that "since they use the variance estimate under a Gaussian assumption of distributions while we use a standard deviation penalization without any distribution assumptions" (Page 1).
    - However, parameterizing a distribution using its first and second moment corresponds to moment matching (which is minimizing the KL divergence between the target distribution and a Gaussian variational distribution.
    - E.g. if the distribution Z would be multi-modal, the standard deviation  and thus \mu - \sigma estimate would not be meaningful

- Applicability: the fact that the parameter must be carefully chosen, not only its specific value but even its order of magnitude (compare
( Hopper-v3 and Walker2d-v3 w) makes it hard to apply in practice. Ideally, the author would have proposed an estimation procedure
to migitate this effect

-  Empirical evaluation:
    - minor things like y-axis labeling on all figures, and unclear what "Mean over 20 trajectories" would mean and no mention of experiment repetitions
    - The only quality metric studied is performance(?) and spread of performance over trajectories. There would be more metrics to estimate robustness, such as worst-case (or some 10,20 percentile) performance. Variation over experiment repetitions etc.



**Summary Of The Paper:**

The paper introduces a risk-sensitive criterion based on the expected value and standard deviation of a distributional RL trianing scheme. The authors evaluate their method w.r.t. robustness over repetitions and find more robust behavior

**Summary Of The Review:**

Overall, I cannot recommend acceptance, because I have concerns regarding the novelty and applicability of the proposed method.

---

> ### Author Response · Authors · 2022-11-16
> **Answer to anYu**
>
> Thank you for all the feedback and all comments you made.
>
> Some comments on the weaknesses you mentioned :
>
> “using a mean/std approach for risk-sensitive RL has been around for some time and is standard procedure with many publications”
>
> Our contribution is in the robustness setting, we show that the mean/std variance approach is also useful in this setting, to our knowledge this is a first.
>
> “E.g. if the distribution Z would be multi-modal, the standard deviation and thus \mu - \sigma estimate would not be meaningful”
>
> For multimodal formulation, the formulation  \mu - \sigma allows taking into account the high variability of the returns of a multimodal distribution to increase the robustness.
>
> “ the fact that the parameter must be carefully chosen, not only its specific value but even its order of magnitude (compare ( Hopper-v3 and Walker2d-v3 w) makes it hard to apply in practice.”
>
> The calibration of the \alpha parameter must be chosen according to the variability of the environment. It is indeed difficult to give a general rule for this parameter theoretically.
>
> Thank you again for all your detailed feedback.

---

> ### Comment · Area_Chair_otJQ · 2022-11-21
> **Any comments to the responses from authors?**
>
> Dear Reviewer anYu,
>
> Thank you very much for your informative review.  The authors have provided responses to your concerns.  How did they change your evaluation, particularly on novelty and clarity?

---

### Official Review · Reviewer_RvL2 · 2022-10-27

**Confidence:** 3
**Correctness:** 3
**Technical Novelty And Significance:** 3
**Empirical Novelty And Significance:** 3
**Recommendation:** 5

**Clarity, Quality, Novelty And Reproducibility:**

There are multiple typos in the current manuscript. The writing needs more polishing. The algorithm described in Section 3.3 is a little bit confusing and hard to read.


**Strength And Weaknesses:**

This paper provides a clear derivation for the reformulation of the robust RL objective to a variance-regularized objective, which is very helpful for readers to understand the motivation. The experimental results seem to be promising and show the effect of regularization on robust performance. However, the experimental comparisons are not complete and thus may not justify the proposed algorithm and its advantage in robust RL compared with other baselines.


**Summary Of The Paper:**

This paper studies risk-averse reinforcement learning problems where the robust objective can be formulated as a penalization term on the standard deviation of the stochastic total return. The authors propose two algorithms for the setting with discrete action space and the setting with continuous action space respectively. Experiments are conducted to compare the proposed method with SAC on Mujoco environments.


**Summary Of The Review:**

As mentioned in the previous sections, I appreciate the mathematical derivation of the reformulation of the robust RL objective. The main shortcoming that I think the authors could improve is its relationship with related methods, which I elaborate in the next paragraph.

There is no comparison with other robust RL methods in the literature. The authors might need to complete this comparison both from an algorithmic design perspective and the experiment perspective. In particular, since there are multiple ideas that motivate the development of the algorithms in this paper such as that of Kumar et al. (2022) and Moskovitz et al. (2021), the authors should at least have some discussion about the performance comparison with them.

“We show that is it possible to improve the Robustness of RL algorithms with variance/standard deviation regularisation”: it is …

Can you explain why the first equality in (1) holds?

In the sentence “...to obtain better gradients when u is small, the (asymmetric) Huber quantile loss”, what do you mean by better gradients?

---

> ### Author Response · Authors · 2022-11-16
> **Answer to RvL2**
>
> Thank you for all the feedback and all comments you did.
>
> You mentioned “there is no comparison with other robust RL methods in the literature”. Indeed, we only compare to SAC which is robust to some uncertainty set. [1]. It could also be interesting to compare to other algorithms such as [2].
>
> “In particular, since there are multiple ideas that motivate the development of the algorithms in this paper such as that of Kumar et al. (2022) and Moskovitz et al. (2021), the authors should at least have some discussion about the performance comparison with them.”
>
> We did not compare to Kumar et al. (2022) as there is no performance on RL continuous  environment, as it is only for MDPs with finite state-action space. Moreover, we said that our performance is similar to Moskovitz et al. (2021) for tested environments on Mujoco in terms of mean performance.
>
> Huber loss is a loss function used in robust regression, that is less sensitive to outliers in data than the squared error loss, and thus usually leads to better performance in the learning of the critic.
>
> Can you explain why the first equality in (1) holds? The full demonstration is explained in Appendix A of the paper.
>
> Thank you again for all your detailed feedback.
>
> [1] B Eysenbach, S Levine - arXiv preprint arXiv:2103.06257, 2021 Maximum entropy rl (provably) solves some robust rl problems.
> [2] Robust Q-learning ,Ashkan Ertefaie, James R. McKay, David Oslin, Robert L. Strawderman

---

> ### Comment · Area_Chair_otJQ · 2022-11-21
> **Any comments to the responses from authors?**
>
> Dear Reviewer RvL2,
>
> Thank you very much for your review.  The authors have provided responses to your concerns.  How did they change your evaluation?

---

### Official Review · Reviewer_pRgU · 2022-10-27

**Confidence:** 5
**Clarity, Quality, Novelty And Reproducibility:** 1/ Novelty

It is not clear to me wha…
**Correctness:** 2
**Technical Novelty And Significance:** 2
**Empirical Novelty And Significance:** 2
**Recommendation:** 3

**Strength And Weaknesses:**

The idea of relating distributional RL for the sake of robustness is interesting and novel, although the connection is quite natural. However, I think there are two major flaws in this paper, in terms of 1/ novelty; 3/ quality. I detail my concerns below.



**Summary Of The Paper:**

This paper aims at achieving robustness in uncertain environments. The proposed approach, a mean-standard deviation formulation, seemingly reduces the desired level of robustness to one parameter tuning. Experiments are conducted on discrete and continuous control domains.

**Summary Of The Review:**

Overall, I think this paper holds an interesting intuition by relating distributional RL to robustness, but it should be further investigated and analyzed: formally, why is the std penalty adding robustness to varying P, theoretically? How different is it from the policy regularization perspective? Why not just do TD learning on the variance as in [2]?

References

[1] Ho, Chin Pang, Marek Petrik, and Wolfram Wiesemann. "Robust Phi-Divergence MDPs." arXiv preprint arXiv:2205.14202 (2022).

[2] Tamar, Aviv, Dotan Di Castro, and Shie Mannor. "Learning the variance of the reward-to-go." The Journal of Machine Learning Research 17.1 (2016): 361-396.

[3] Pan, Xinlei, et al. "Risk averse robust adversarial reinforcement learning." 2019 International Conference on Robotics and Automation (ICRA). IEEE, 2019.

[4] Wiesemann, Wolfram, Daniel Kuhn, and Berç Rustem. "Robust Markov decision processes." Mathematics of Operations Research 38.1 (2013): 153-183.

[5] Puterman, Martin L. "Markov decision processes." Handbooks in operations research and management science 2 (1990): 331-434.

[6] Derman, Esther, Matthieu Geist, and Shie Mannor. "Twice regularized MDPs and the equivalence between robustness and regularization." Advances in Neural Information Processing Systems 34 (2021): 22274-22287.

---

> ### Author Response · Authors · 2022-11-16
> **Answer to pRgU**
>
> Thank you for all the feedback and all comments you made.
>
> Answers on novelty :
>
> Paper [1] deals with the case of f-divergence in effect, it is not published and from 2022 this is why we were not aware of this work. Moreover, paper [1] deals with the case of f-divergence indeed but only for MDPs with action space and finite state, It does not look at the robustness effect but only at the execution time of the algorithm. Our work attempts to obtain a close form solution using regularization in order to be able to adapt the robustness to large-scale and continuous RL problems.
>
> Our approach is different from  [3] in that it is theoretically justified compared to the paper, that the variance used in this paper is that through different models and not that of the distribution of returns. Although the formulation is close, it is not theoretically justified.
> Moreover, their algorithm is not on a classical RL benchmark and the algorithm only deals with the discrete case. Furthermore, it requires interacting with several models, while our approach requires only one.
>
> The contribution comes from the fact that we do robustness on the whole trajectory, which makes it possible to introduce the distribution of returns and the use of the distributional RL. It should be noted that this is the first work to address this type of robustness from our knowledge. This is a significant contribution that sets us apart from other work. Experimentally, it is shown that this improves stability over a range where the environment does not change too much.
>
> Policy regularization is not the same idea even if it can be robust to certain uncertainty sets, as it is discussed in this paper [7] cited in our work. A penalization in relation to the variability of the returns allows robustness in the dynamics [6].
>
> About Quality/Clarity :
>
> "The 3rd § of Sec 1 is unclear to me. The authors claim that they focus on more general, continuous state space, but they do not provide any theoretical result. Additionally, although not formally studied in most RL works, most standard properties such as contractive operator, greediness, etc., still hold for continuous action and state spaces, as long as the action space is compact and the state space Polish + measurable (see [5])"
>
> Here we focus on a more general continuous state space S with a discrete or continuous action space A and with constraints defined using f -divergence. Thank you for the remark. We were indeed thinking of this classical extension. In particular, we propose an algorithm which can be used in a continuous state/continuous action world provided the argmax can be computed.
>
> "Why not just do TD learning on the variance as in [2]? "
> Using TD learning would be possible, but it appears less stable in practice. We also want to use the distributional RL in order to have state-of-the-art performance in terms of mean performance.
>
> Thank you again for all your detailed feedback.
>
> [1] Ho, Chin Pang, Marek Petrik, and Wolfram Wiesemann. "Robust Phi-Divergence MDPs." arXiv preprint arXiv:2205.14202 (2022).
> [2] Tamar, Aviv, Dotan Di Castro, and Shie Mannor. "Learning the variance of the reward-to-go." The Journal of Machine Learning Research 17.1 (2016): 361-396.
> [3] Pan, Xinlei, et al. "Risk averse robust adversarial reinforcement learning." 2019 International Conference on Robotics and Automation (ICRA). IEEE, 2019.
> [4] Wiesemann, Wolfram, Daniel Kuhn, and Berç Rustem. "Robust Markov decision processes." Mathematics of Operations Research 38.1 (2013): 153-183.
> [6] Derman, Esther, Matthieu Geist, and Shie Mannor. "Twice regularized MDPs and the equivalence between robustness and regularization." Advances in Neural Information Processing Systems 34 (2021): 22274-22287.
> [7] B Eysenbach, S Levine - arXiv preprint arXiv:2103.06257, 2021 Maximum entropy rl (provably) solves some robust rl problems.

---

> ### Comment · Area_Chair_otJQ · 2022-11-21
> **Any comments to the responses from authors?**
>
> Reviewer pRgU,
>
> Thank you very much for your detailed review.  The authors have provided responses to your concerns.  How did they change your evaluation, particularly on novelty?

---

### Decision · Program_Chairs · 2023-01-20

**Decision:**

Reject

**Justification For Why Not Higher Score:**

The proposed approach has limited novelty, and its empirical advantages over existing methods are unclear.

**Justification For Why Not Lower Score:**

N/A

**Metareview: Summary, Strengths And Weaknesses:**

This paper proposes a reinforcement learning (RL) algorithm that is robust against uncertainties in the environment.  The proposed approach is well theoretically grounded with clean derivation, which constitute the main strength of the paper.

Major concerns are in the limited novelty and practical advantages over existing robust or risk-sensitive RL methods.  The proposed approach is essentially the mean -  standard deviation formulation, which has been studied extensively in the literature.  The experiments only compare the proposed approach against SAC and PPO, and the empirical advantages of the proposed approach to existing robust or risk-sensitive RL methods are unclear.